# Determining Symptomatic Factors of Nomophobia in Peruvian Students from the National University of Engineering

**Jimmy Aurelio Rosales-Huamani** [1,*,†] , **Jose Luis Castillo-Sequera** [2,†] , **Rita Rocio Guzman-Lopez** [1,†] , **Eder Eliseo Aroni-Vilca** [1,†] and **Carmen Rosalia Matos-Avalos** [1,†]

1   National University of Engineering, Lima 15333, Peru; rrguzman@uni.edu.pe (R.R.G.-L.); ederaronivilca@gmail.com (E.E.A.-V.); carmatos@uni.edu.pe (C.R.M.-A.)
2   Department of Computer Science, Higher Polytechnic School, University of Alcala, 28871 Alcalá de Henares, Spain; jluis.castillo@uah.es
*   Correspondence: jrosales@uni.edu.pe; Tel.: +51-1-381-5630
†   These authors contributed equally to this work.

**Abstract:** The use of cell phones has increased worldwide in the past few decades, particularly in children and adolescents. Using these electronic devices provides personal benefits. Communicating through cell phones was a very important factor in the socioeconomic progress of developed countries. However, it is beyond doubt that its indiscriminate use can bring up certain psychiatric disorders or cause some disorder in a person, within the phobic group of anxiety disorders called nomophobia; basically associated with anxiety, nervousness, discomfort, and distress when contact with the smartphone is lost, mainly in the youngest users. This research proposal aims to identify symptoms that have not yet been detected by unceasing cell phone use, considering that in Peru there are few studies of human health engineering and the physical mental health. For that reason, we sought to identify the symptomatic factors of nomophobia presented by students at the National University of Engineering and its interference with their academic life. To accomplish this study, we designed a questionnaire according to our reality with the use of focus groups techniques when the test was taken in class. Three symptomatic factors of nomophobia were identified: feelings of anxiety, compulsive smartphone use, and feelings of anxiety and panic. The study included a representative sample of 461 students in different years of study engineering (21% women, 79% men, over 17 years of age). Finally, given the widespread adoption of smartphones and their integration into educational environments, the results of this study can help educators understand students' inclination to use their smartphones at all times.

**Keywords:** nomophobia; anxiety; smartphone; internet; cyberaddiction; new technologies

---

## 1. Introduction

Continuous technological developments have changed the way that human beings manage their daily activities. In particular, information and communication technologies have became an indispensable part of our social interactions, work activities, and education.

However, the appearance of smaller electronic devices with higher computing capabilities has enabled the proliferation of cheap mobile devices and smartphones are considered to be the latest Information and Communications Technology (ICT) development [1]. They are no longer just cell phones as, in addition to making calls or sending messages, they enable access to the Internet and thus a wide variety of services offered by the network [2]. Cell phone use is widespread, even surpassing

the population in some countries [3], while 70% of young people (15–24 years old) in the world use the Internet through various means, including mobile devices [4]. They are used more widely among the youngest population. This is because young people have adopted new technologies quickly and because using a smartphone is a status symbol within the tech-culture [5]. At the end of 2017 in Peru, the number of mobile lines that access the Internet was 21.2 million, this means that two of every three people accessed the Internet through mobile services, which gives us an idea of the number of smartphones in Peru [4]. On the other hand, social networks have became one of the largest and most influential components on the web, providing an easy platform to everyone (young and the elderly alike). Many young people lose sight of the real world as they are absorbed by the virtual world and become slaves of technology [6].

Despite their considerable positive impacts, smartphones have a seemingly minor negative impact, called nomophobia, which in turn can be as serious as the positive side if these phones are not wisely used [7]. Nomophobia is an abbreviation of the English "no mobile phone phobia", which translates into the fear of not having a cell phone. Specific phobias are frequent anxiety disorders; at the same time, they precede other psychiatric disorders such as depression and abusive use of toxic substances. There have been problematic situations where people are too close to their smartphones, presenting symptoms of behavioral addiction that interferes with their daily activities.

Studies have also been conducted on the problem of indiscriminate cell phone use. In Walsh, White and Young presented a preliminary examination of the behavior of young Australians and their use of cell phones. This study explored the relationship between psychological predictors of the frequency of cell phone use and the participation of cell phones conceptualized as a cognitive and behavioral interaction of people with their cell phones. The participants in this study were young Australians between 15 and 24 years of age [8]. Independently, King et al. presented a study applied to individuals with panic disorder and agoraphobia because of dependence on their cell phones [9]. Then, Krajewska et al. examined the role of cell phones in the lives of Belorussian and Polish students. This study included the analysis of a sample of students from Belarus and Poland. Consequent to this study, they concluded that almost 20% of students in Poland and 10% in Belarus have symptoms of cell phone addiction [10]. Independently, Ouslasvirta et al. mentioned that smartphones may cause compulsive checking habits [1].

King et al. argued that studies on the relationships between individuals and new technologies are relevant with the justification that new technologies produce changes in behavior, as well as feelings and symptoms, that must be studied and monitored continuously in modern society [11]. Then, Castilla and Paez, among their research results, mentioned that the most used application among young people is "WhatsApp" for instant messaging and that they may feel social exclusion if they are not included in a group [12].

Billieux et al. mentioned that, despite the many positive results, cell phones when used excessively are now often associated with potentially harmful and/or disruptive behaviors [13]. Independently, Yildirim and Correia considered nomophobia to be the phobia of the modern age that has been introduced into our lives as a by-product of the interaction between people and mobile information and communication technologies, particularly smartphones. In their studies, the authors have identified and described the dimensions of nomophobia and developed a questionnaire to measure this nomophobia [14].

The definition of nomophobia is controversial [5]. Therefore, it is necessary to define the one that will be used for this study. We consider nomophobia to be a behavioral addiction to cell phones, which is manifested by psychological and physical symptoms of dependence. Yildirim and Correia mentioned that nomophobia is an abbreviated form of "no mobile phone phobia" and is thought to stem from the excessive use of a mobile phone [14]. Independently, Arpaci mentioned that nomophobia is a specific disorder caused by smartphone use [15]. Then, Dasgupta et al. mentioned that the purpose of their study is to find out the predominance of nomophobia among the smartphones used by undergraduates of West Bengal [16]. In their work, they used the Nomophobia Questionnaire (NMP-Q) developed by Yildirim and Correia [14].

The teaching of each course at the National University of Engineering requires on average 2 to 3 h; however, there are many factors that can cause distraction in the classroom. One factor is the excessive use of cell phones. Aguilera et al. presented a study to analyze the relationship between the level of nomophobia and the distraction associated with the use of smartphones. The study population were nursing students from the University of Almeria in Spain [17].

Froese et al. examined the impact of smartphone use in classroom learning; for this, they give a test to the students at the end of class, proving that those who had the lowest score were students that immediately answered the text messages from their cell phone with regard to who had kept their cell phone [18]. Independently, the studies of [19,20] mentioned that the ringing or notification of the cell phone, regardless of phone number, can be a distractor in the classroom. Then, Prasad et al. mentioned that the objective of the study is to evaluate the pattern of usage of mobile phones and its effects on the academic performance of dental students in India College. The authors concluded that there is an alarming indication of mobile phone addiction on the part of students, which affects their academic performance in a negative way [21]. Arpaci et al. presented a study that aims to investigate the mediating effect of mindfulness on the relationship between attachment and nomophobia. The study is also focused on gender differences in attachment, mindfulness, and nomophobia [22].

Mendoza et al. mentioned that the excessive use of cell phones has led researchers to focus on how the use of cell phones affects learning and memory in classroom. The participants were recruited from undergraduate psychology courses [23]. Independently, Bychkov and Young mentioned that there are symptoms of nomophobia from excessive use of the smartphone. Fortunately, these symptoms may be resolved through behavioral, as the technology itself may be useful tool for that very behavior therapy—for example, transmitting health ads published in radio, television, text messages, and mobile apps. From this research, the authors perform a review systematics of applications for mobiles for the decrease of the addiction to smartphones [24]. Lee et al. examined the relationship between the Nomophobia Questionnaire (NMP-Q) and the Obsessive Content Scale (OBS). This study helps to better understand personality disorders (e.g., obsessiveness) that are emerging from the excessive use of mobile phones or the excessive fear of losing a cell phone [25]. Independently, Balshan et al. developed and validated the Arabic version of the nomophobia questionnaire in university students of Kuwait. In their study, the authors concluded that more than half of the sample contained a moderate nomophobia and a quarter of the sample contained a serious level of nomophobia [26].

Then, Ahmed et al. carried out an online cross-sectional survey that was conducted by using the Google form platform. This research involved a total of 157 students in the course of physiotherapy at the University of North India, and used the questionnaire NMP-Q [27]. Ayar et al. analyzed levels of nomophobia in which the participants were undergraduate nursing students in a school located in western Turkey. In this study, the authors used several models to correlate the effect of problematic internet use, social appearance anxiety, and social media with the nomophobia [28]. Then, Gutierrez-Puertas et al. mentioned that the objective of this study was to compare the levels of nomophobia, experienced by nursing students at the University of Almeria, Spain and the Polytechnic Institute of Braganza, Portugal. The conclusions found by the authors indicate that the dimensions explored contain significant levels of nomophobia between both populations of nursing students [29].

On the other hand, the use of mobile phones causes alterations in the habits of daily life and perceptions of reality, which can be associated with negative results, such as deteriorated social interactions, social isolation, and mental health problems such as anxiety, depression, and stress. The present study discusses nomophobia in relation to the smartphone. In the absence of research information on nomophobia in our country, we believe that, by carrying out this study, we will be able to identify the main symptomatic factors that are presented by the National University of Engineering (UNI) students that are caused by indiscriminate cell phone usage, and that are not observable during the course of the students' daily lives. To identify these factors, our contribution has been to give viability to the test adapted with the objective of getting an answer according to our reality, and the use of focus group techniques. Consequently, we present three symptomatic factors

of nomophobia in UNI students, which describe the need that students have for indiscriminate cell phone use.

The present study is described as follows: In Section 2, we explain the methodology to be used. In Section 3, we describe the results obtained. Finally, in Section 4, we show the conclusions and future studies.

## 2. Methodology

We have made a transversal, correlational, and factorial research methodology. It was carried out in 461 male and female students at the National University of Engineering. To carry out the proposal, we completed the following stages that are shown in Figure 1, trying to adapt a methodology that allows for covering the most relevant aspect of our study and following the recommendations of other authors.

- Test Adaptation. Currently, there is a wide variety of tools to identify dependency problems that are associated with smartphone use and the use of other information and communication technologies. However, in our study, we used the "Test of Mobile Phone Dependence (TMDbrief) Questionnaire" [30] as a main reference, which evaluates the main characteristics of cell phone dependence: tolerance, withdrawal syndrome, change of impulse control, excessive use, etc., using an intercultural approach. In our study, the test was improved through the methodology of focus group. This methodology was used as a tool to analyze the initial questionnaire and collect the general recommendations of many students from different careers. The meeting was moderated by the principal investigator. After the meeting, we concluded that some of the survey questions were improved in our research. There are several authors [31,32] who use the focus group techniques.
- Test Validation. In this stage, the questionnaire was validated using the Cronbach test [33].
- Sample Methodology. In this stage, the sample methodology was defined to calculate sample size in correlational studies and is proportional to the population of the faculties in the University.
- Factor Analysis Application. Taking into account the objective of this study, we are interested in finding out whether questions in part 2 (below) in the test is joined with some special feature or not. For this reason, we have decided to applicate factorial analyses. Thus, we looked for a smaller number of factors by reduction of variables. By the way, the minimum factors could be able to explain information inside data.
- Experimental Results. The results were processed using Statistical Package for the Social Sciences (SPSS) software for Windows (IBM Corp. Released 2013. IBM SPSS Statistics for Windows, Version 22.0. Armonk, NY, USA) to obtain the final reports and to be able to obtain the respective conclusions.

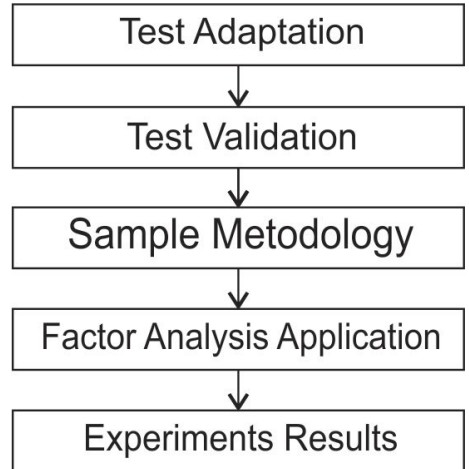

**Figure 1.** Methodology to carry out the proposal.

*2.1. Quantitative Instrument*

The test used was carefully adapted and translated from the Test of Mobile Phone Dependence (TMDbrief) [30] on the basis of previous recommendations and previous study experience [2]. Because of the multicultural approach of TMDbrief, it was not necessary to validate the translation.

Moreover, the test applied was objective and consists of two blocks. The first block corresponds to basic demographic data. The second block has 16 items, which show how UNI students relate to their smartphones, and each item proposes a statement that has seven response options, a Likert scale, with respect to students' level of agreement or disagreement with each sentence: 1 = Strongly disagree to 7 = Strongly agree. This is shown in the following survey.

NATIONAL UNIVERSITY OF ENGINEERING

We want to adapt and apply the following test to the UNI student population, with the goal of offering help to those who use their cell phones in an unhealthy manner. Thank you for your responses.

1. Age
2. Gender:                 a. Female                 b. Male
3. Department                              4. Major
5. Cycle                                   6. Origin

Answer or mark with "x", belongs to your opinion:

7. How long (in years) have you been use your smartphone or cell-phone:
   a) Lees one year
   b) More one year, but less two years        c) More two years, but less three years
   d) More three years, but less four years     e) More four years, but less five years
   f) More five years

8. Do you have a plan of mobile data that allows you to accessing to the internet through your smartphone or cell-phone?
   a. yes                  b. not

9. Approximately, how much time during the day do you think that you use your cell phone or smartphone? .... hours.

10. More and less, how much time during the day do you check your smartphone or cell-phone? .... times.

11. How frequently do you think that you check your smartphone or cell-phone?
   a) Each 5 minutes
   b) Each 10 minutes                          c) Each 20 minutes
   d) Each 30 minutes                          e) Each hour
   f) Others (specify please)

## Block 1

The following questions related to how you use your cell phone:

| Please indicate your level of agreement or disagreement with an "x" for the following statements in relation to smartphone or cell phone use | 1=strongly disagree | | | | 7=strongly agree | | |
|---|---|---|---|---|---|---|---|
| 1. If my smartphone or cell phone wasn't working for a long time and it would take a long time to fix it, I would feel very bad. | 1 | 2 | 3 | 4 | 5 | 6 | 7 |
| 2. If I don't have my smartphone or cell phone, I feel bad. | | | | | | | |
| 3. I don't think I could handle a week without my smartphone or cell phone. | | | | | | | |
| 4. If I couldn't check my smartphone or cell phone for a while, I would want to check it. | | | | | | | |
| 5. I spend more time than I should talking on my smartphone or cell phone, sending messages, and using other apps. | | | | | | | |
| 6. I go to bed later or have slept less to use my smartphone or phone cell. | | | | | | | |
| 7. I use my smartphone or cell phone (for calls, reading or sending messages, using WhatsApp, among other things) in situation that may not be dangerous but are not appropriate for smartphone use (such as while others are talking to me, etc). | | | | | | | |
| 8. I need to use my smartphone or cell phone more often. | | | | | | | |
| 9. I get angry or irritated when someone bothers me while I'm using my smartphone or cell phone. | | | | | | | |
| 10. If my smartphone or cell phone is with me, I can't stop using it. | | | | | | | |
| 11. Ever since I have had my smartphone or cell phone, I have increased sent messages. | | | | | | | |
| 12. As soon as I get up in the morning, the first thing I do is see who called up or whether anyone sent me a message. | | | | | | | |
| 13. When I feel lonely, I use my smartphone or cell phone to make calls, send messages, etc. | | | | | | | |
| 14. Right now, I would grab my smartphone or cell phone and send message, make a call, or check social networks. | | | | | | | |
| 15. I feel nervous if I do not receive messages, calls, and notifications from social networks on my smartphone or cell phone. | | | | | | | |
| 16. If I didn't have my smartphone or cell phone with me, I would feel bad because I wouldn't be able to check social networks. | | | | | | | |

Block 2

2.1.1. Validation of the Instrument

The reliability analysis was carried out to understand the internal consistency of the scale, i.e., the correlation between the items analyzed, as well as to assess the reliability or homogeneity of the questions [33] in block 2. Cronbach's alpha coefficient ($\alpha$) oscillates between 0 and 1, where 0 means a reliability assessment of null and 1 represents total reliability. In addition, internal consistency is

considered high if it is between 0.70 and 0.90. Values below 0.70 indicate low internal consistency and those above 0.90 suggest that the scale has several items that measure exactly the same [34].

To calculate Cronbach's alpha coefficient, using the variance of the items and the variance of the total score, we use the following formula:

$$\alpha = \left[\frac{k}{k-1}\right]\left[1 - \frac{\sum\limits_{i=1}^{k} S_i^2}{S_T^2}\right], \tag{1}$$

where:

$S_i^2$　:　Is the variance of each item,
$S_T^2$　:　Is the variance of all rows,
$k$　:　Is the number of questions or items.

The calculation of the value of Cronbach's alpha was processed with the help of SPSS Software. Table 1 shows the results.

**Table 1.** Reliability statistics.

| Cronbach's Alpha | Cronbach's Alpha Based on the Categorized Elements | Number of Elements |
|:---:|:---:|:---:|
| 0.873 | 0.883 | 16 |

The value of Cronbach's alpha ($\alpha = 0.873$) shows that the questionnaire exhibits high internal consistency. The questionnaire was previously improved in focus group sessions to verify the interpretation and adequacy of the items.

Malhotra defined the pilot test as applying a questionnaire to a small sample of the units of analysis to identify and eliminate possible problems in the questionnaire's design [35]. The instrument in question was validated with a pilot sample of 30 students from the National University of Engineering to eliminate inconsistencies or questions within the questionnaire.

*2.2. Sample Size*

It is necessary to estimate the correlations, relationship, or association between the two variables–symptoms and indiscriminate smartphone use—hence, it is necessary to establish the calculation of the sample size using the following formula [36]:

$$n_0 = \left\{\frac{Z_{1-\alpha/2} + Z_{1-\beta}}{\frac{1}{2} \cdot \ln\frac{1+r}{1-r}}\right\}^2 + 3, \tag{2}$$

where:

$n_0$　:　Sample size,
$\alpha$　:　Level of significance, which is universally chosen as 5% (error type I),
$Z_{1-\alpha/2}$　:　Value of the standard normal variable corresponding to a confidence level,
$\beta$　:　Probability of accepting a false hypothesis, when this is really false (error type II), this value is fixed around 0.2 in a majority of cases, thus it will have a test power of 80%,
$Z_{1-\beta}$　:　P-Normal variable value for a test power of 85%, the value of which in the normal table,
$r$　:　is 1.04 Value of the correlation from which a relationship is considered in our study.

Assuming a 5% level of significance, a test power of 85%, and $r = 0.15$, a sample size of 397 students is reached. This size was increased to 16% of the size calculated to cover the non-response

rate, culminating in the obtainment of a sample of 461 students to be used. In this way, we comply with evaluating a representative sample for our study, which guarantees valid results.

*2.3. Variable Classification*

Variables: Age, Gender, Department, and Major correspond to variables for the descriptive study of the sample (block 1). The dependent variables, i.e., the scores obtained from the 16 statements on how students relate to their smartphones (indiscriminate smartphone use), and the independent variable, symptoms, are detailed in 16 sentences in the second block of the previous survey.

*2.4. Data Collection Procedures*

The questionnaires were administered to the students on the National University of Engineering campus at different times and in areas near the university, the participants had 10 min to complete the test. We tried to consider different schedules because first semester students have classes in the mornings, while those in the last cycles usually have classes in the afternoon or evening. The respondents answered the questionnaire's questions freely and voluntarily. The recommendation was to respond truthfully and to try to answer as quickly as possible. No incentive was offered for participation. With this, it was possible to collect the data in an anonymous and reliable way, covering different types of students of the university.

## 3. Results and Discussion

*3.1. Descriptive Analysis*

A total of 461 completed questionnaires were processed from the database, using the SPSS software package for Windows (version 19.0, SPSS, Inc., Chicago, IL, USA). According to reports issued by SPSS, the sample consisted of 21% men and 79% women. With regard to the ages of the students in the sample, 35.8% were 17–19 years old; 30.4% were 20–21 years old; and 33.8% were 22 years old and above. The mean and standard deviation of age was $20.81 \pm 0.12$ with an age range of 17 to 32 years. See Table 2 for these results.

**Table 2.** Percentage distribution of respondents by age.

| Age | Frequency | Percentage (%) |
|:---:|:---:|:---:|
| [17, 19] | 165 | 35.8 |
| [20, 21] | 140 | 30.4 |
| [22, or more] | 156 | 33.8 |
| **Total** | **461** | **100** |

With regard to how long they had used a smartphone, 31.9% of students had one in use for more than five years; 30.2% had one in use for 3–4 years; and 37.9% had one in use for less than two years. See Table 3 for these results.

**Table 3.** How long the respondent has had a smartphone.

| | Frequency | Percentage (%) |
|:---|:---:|:---:|
| Less than a year | 37 | 8 |
| More than 1 year but less than 2 | 72 | 15.6 |
| More than 2 years but less than 3 | 66 | 14.3 |
| More than 3 years but less than 4 | 77 | 16.7 |
| More than 4 years but less than 5 | 62 | 13.5 |
| More than 5 years | 147 | 31.9 |
| **Total** | **461** | **100** |

When asked whether the students had a data plan that would allow them to access the Internet, 68.3% of the respondents answered yes and 31.7% answered that they did not have a data plan.

With regard to the total time dedicated to smartphone use per day, 26.8% answered that they used their smartphones for a total of 1–3 h a day. The majority (34.1%) answered that they used their smartphones for 4–5 h; 19.1% answered saying 5–10 h; and 20% answered saying 10 or more hours a day. Table 4 shows these results.

**Table 4.** Time per day devoted to smartphone use.

| Time per Day | Frequency | Percentage (%) |
|:---:|:---:|:---:|
| 1 to 3 h | 124 | 26.8 |
| 4 to 5 h | 157 | 34.1 |
| 6 to 9 h | 88 | 19.1 |
| 10 or more | 92 | 20.0 |
| **Total** | **461** | **100** |

With regard to the number of times they usually checked their smartphones in a day, 25.8% of the respondents answered that they checked 1–8 times; 24.5% checked 9–16 times; 29.7% checked 17–30 times; and 20% checked 31 or more times. Table 5 shows these results.

**Table 5.** Frequency of checking smartphone or cell phone per day.

| Time per Day | Frequency | Percentage (%) |
|:---:|:---:|:---:|
| 1 to 8 times | 119 | 25.8 |
| 9 to 16 times | 113 | 24.5 |
| 17 to 30 times | 137 | 29.7 |
| 31 or more times | 92 | 20.0 |
| **Total** | **461** | **100** |

We can note that 7.6% of students responded to the survey stating that they checked their smartphones every 5 min; 33.2% checked every 10–20 min; 38.2% checked every 30–60 min; and 21% checked every 2 h or less. Table 6 shows these results.

**Table 6.** How often do you think you usually check your smartphone or cell phone?.

| Frequency | Frequency | Percentage (%) |
|:---:|:---:|:---:|
| Every 5 min | 35 | 7.6 |
| Every 10 min | 76 | 16.5 |
| Every 20 min | 77 | 16.7 |
| Every 30 min | 93 | 20.2 |
| Every hour | 83 | 18.0 |
| Every 2 h | 37 | 8.0 |
| Every 3 h or less | 60 | 13.0 |
| **Total** | **461** | **100** |

### 3.2. Correlation Analysis

In the item correlation matrix of block 2, we observed variables that correlate moderately, with the rest of the variables exhibiting low correlations. However, the determinant value of the correlation matrix is close to 0, which indicates that the matrix variables are linearly related, which, in turn, supports the continuity of the analysis in the main components. See Table 7 for these results.



**Table 7.** Correlation matrix for the 16 items in the questionnaire.

| Items | Q-1 | Q-2 | Q-3 | Q-4 | Q-5 | Q-6 | Q-7 | Q-8 | Q-9 | Q-10 | Q-11 | Q-12 | Q-13 | Q-14 | Q-15 | Q-16 |
|---|---|---|---|---|---|---|---|---|---|---|---|---|---|---|---|---|
| Q-1 | 1.000 | 0.600 | 0.482 | 0.408 | 0.271 | 0.273 | 0.190 | 0.292 | 0.229 | 0.276 | 0.213 | 0.234 | 0.244 | 0.268 | 0.261 | 0.367 |
| Q-2 | 0.600 | 1.000 | 0.507 | 0.457 | 0.313 | 0.258 | 0.236 | 0.402 | 0.406 | 0.400 | 0.247 | 0.252 | 0.302 | 0.343 | 0.435 | 0.480 |
| Q-3 | 0.482 | 0.507 | 1.000 | 0.450 | 0.336 | 0.239 | 0.196 | 0.365 | 0.450 | 0.404 | 0.206 | 0.162 | 0.179 | 0.307 | 0.419 | 0.406 |
| Q-4 | 0.408 | 0.457 | 0.450 | 1.000 | 0.374 | 0.390 | 0.261 | 0.306 | 0.273 | 0.369 | 0.281 | 0.356 | 0.356 | 0.316 | 0.313 | 0.445 |
| Q-5 | 0.271 | 0.313 | 0.336 | 0.374 | 1.000 | 0.475 | 0.350 | 0.351 | 0.286 | 0.427 | 0.320 | 0.324 | 0.334 | 0.378 | 0.335 | 0.345 |
| Q-6 | 0.273 | 0.258 | 0.239 | 0.390 | 0.475 | 1.000 | 0.316 | 0.292 | 0.181 | 0.334 | 0.302 | 0.366 | 0.320 | 0.275 | 0.219 | 0.283 |
| Q-7 | 0.190 | 0.236 | 0.196 | 0.261 | 0.350 | 0.316 | 1.000 | 0.332 | 0.241 | 0.255 | 0.193 | 0.277 | 0.288 | 0.328 | 0.274 | 0.313 |
| Q-8 | 0.292 | 0.402 | 0.365 | 0.306 | 0.351 | 0.292 | 0.332 | 1.000 | 0.396 | 0.457 | 0.211 | 0.262 | 0.254 | 0.452 | 0.437 | 0.415 |
| Q-9 | 0.229 | 0.406 | 0.450 | 0.273 | 0.286 | 0.181 | 0.241 | 0.396 | 1.000 | 0.533 | 0.211 | 0.169 | 0.262 | 0.417 | 0.606 | 0.506 |
| Q-10 | 0.276 | 0.400 | 0.404 | 0.369 | 0.427 | 0.334 | 0.255 | 0.457 | 0.533 | 1.000 | 0.198 | 0.341 | 0.295 | 0.472 | 0.495 | 0.479 |
| Q-11 | 0.213 | 0.247 | 0.206 | 0.281 | 0.320 | 0.302 | 0.193 | 0.211 | 0.211 | 0.198 | 1.000 | 0.269 | 0.394 | 0.172 | 252 | 263 |
| Q-12 | 0.234 | 0.252 | 0.162 | 0.356 | 0.324 | 0.366 | 0.277 | 0.262 | 0.169 | 0.341 | 0.269 | 1.000 | 0.426 | 0.371 | 0.234 | 0.369 |
| Q-13 | 0.244 | 0.302 | 0.179 | 0.356 | 0.334 | 0.320 | 0.288 | 0.254 | 0.262 | 0.295 | 0.394 | 0.426 | 1.000 | 0.314 | 0.313 | 0.321 |
| Q-14 | 0.268 | 0.343 | 0.307 | 0.316 | 0.378 | 0.275 | 0.328 | 0.452 | 0.417 | 0.472 | 0.172 | 0.371 | 0.314 | 1.000 | 0.469 | 0.529 |
| Q-15 | 0.261 | 0.435 | 0.419 | 0.313 | 0.335 | 0.219 | 0.274 | 0.437 | 0.606 | 0.495 | 0.252 | 0.234 | 0.313 | 0.469 | 1.000 | 0.601 |
| Q-16 | 0.367 | 0.480 | 0.406 | 0.445 | 0.345 | 0.283 | 0.313 | 0.415 | 0.506 | 0.479 | 0.263 | 0.369 | 0.321 | 0.529 | 0.601 | 1.000 |

### 3.3. Kaiser–Meyer–Olkin (KMO) Measurement

This indicates the percentage of variance that the analyzed variables have in common; 0.6 and above is considered a good sample adaptation for a factor analysis [37,38]:

$$KMO = \frac{\sum_{i \neq j} r_{ij}^2}{\sum_{i \neq j} r_{ij}^2 + \sum_{i \neq j} r_{ij,m}^2},$$

(3)

where:

$r_{ij}$: Represents the simple correlation coefficient between the variables *i* and *j*.

$r_{ij,m}$: Represents the partial correlation coefficient between the variables *i* and *j*, eliminating the effect of the remaining m variables.

In Table 8, we see that the Kaiser–Meyer–Olkin (KMO) that was obtained, take the value = 0.913 > 0.6; hence, it indicates that the data reduction process is good.

**Table 8.** Kaiser–Meyer–Olkin and Bartlett Test.

| Kaiser–Meyer–Olkin sampling adequacy measurement | | 0.913 |
|---|---|---|
| Barlett's sphericity test | Approx. Chi squared | 2748.056 |
| | df | 120 |
| | Sig. | 0.000 |

### 3.4. Bartlett's Sphericity Test

The Bartlett sphericity test result that contrasts its null hypothesis that the correlation matrix is an identity matrix (there is no correlation between the variables) has been obtained as shown in Table 8 (*p*-value = 0.000 < 0.05); hence, the Bartlett null hypothesis is rejected. Thus, the results of these tests indicate that the factor analysis can be considered to be appropriate [39–41].

### 3.5. Factor Analysis

In Table 9, which shows explained variance; three factors explain 55.166% of the variance. These factors are extracted via the analysis of main components, and the criteria that support its application are the Kaiser–Meyer–Olkin measurement test result, which takes a value of 0.913, and the Bartlett sphericity test, in which the *p*-value < 0.05; hence, it makes sense to perform the factor analysis.

**Table 9.** Total explained variance. Extraction method: Principal component analysis.

| Component | Initial Eigenvalues | | | Extraction Sums of Squared Loadings | | | Rotation Sums of Squared Loadings | | |
|---|---|---|---|---|---|---|---|---|---|
| | Total | % Variance | % Gathered | Total | % Variance | % Gathered | Total | % Variance | % Gathered |
| 1 | 6.187 | 38.668 | 38.668 | 6.187 | 38.668 | 38.668 | 3.517 | 21.984 | 21.984 |
| 2 | 1.453 | 9.084 | 47.752 | 1.453 | 9.085 | 47.752 | 2.917 | 18.231 | 40.214 |
| 3 | 1.186 | 7.414 | 55.166 | 1.186 | 7.414 | 55.166 | 2.392 | 14.952 | 55.166 |

With regard to the component matrix, to be able to perform the interpretation of the factors, we used Table 10 on the rotated component matrix by rotating varimax [39–41] to discover hidden relationships within the components and the respective indicators, which facilitates the interpretability of the factors. The table highlights values above 0.45 to achieve better exposure of the initial variables obtained for each component or factor.

**Table 10.** Rotated component matrix and its associated indicators.

| Item | | Component | | |
|---|---|---|---|---|
| | | 1 | 2 | 3 |
| Q9 | I get angry or irritated when someone bothers me while I'm using my smartphone. | **0.784** | 0.039 | 0.186 |
| Q15 | I feel nervous if I do not receive messages, calls, and notifications from social networks on my smartphone. | **0.783** | 0.146 | 0.201 |
| Q10 | If my smartphone is with me, I can't stop using it. | **0.678** | 0.264 | 0.206 |
| Q16 | If I didn't have my smartphone with me, I would feel bad because I wouldn't be able to check social networks. | **0.666** | 0.271 | 0.302 |
| Q14 | Right now, I would grab my smartphone and send a message, make a call, or check social networks. | **0.658** | 0.339 | 0.074 |
| Q8 | I need to use my smartphone more often. | **0.585** | 0.243 | 0.212 |
| Q12 | As soon as I wake up in the morning, the first thing I do is see who called me or whether someone has sent me a message. | 0.198 | **0.689** | 0.057 |
| Q6 | I go to bed later or have slept less to use my smartphone. | 0.090 | **0.684** | 0.215 |
| Q13 | When I feel lonely, I use my smartphone to make calls, send messages, etc. | 0.191 | **0.667** | 0.100 |
| Q5 | I spend more time than I should talking on my smartphone, sending messages, and using other apps. | 0.295 | **0.581** | 0.197 |
| Q11 | Ever since I have had smartphone, I have increased sent messages. | 0.059 | **0.560** | 0.207 |
| Q7 | I use my smartphone (for calls, reading, or sending messages and using WhatsApp, among other things), in situations that may not be dangerous but are not appropriate for smartphone use (such as while others are talking to me, etc.). | 0.326 | **0.495** | 0.007 |
| Q1 | If my smartphone wasn't working for a long time and it would take a long time to fix it, I would feel very bad | 0.104 | 0.219 | **0.809** |
| Q2 | If I don't have my smartphone, I feel bad. | 0.373 | 0.176 | **0.731** |
| Q3 | I don't think I could handle a week without my smartphone. | 0.393 | 0.049 | **0.692** |
| Q4 | If I couldn't check my smartphone for a while, I would want to check it. | 0.166 | 0.461 | **0.570** |

Extraction method: analysis of main components. Rotation method: Varimax with Kaiser standardization.
a: The rotation converged in six iterations.

The matrix shows three components, where each component has 16 items, and on the basis of Table 10, we can interpret each of them:

**Component 1:** This component includes the set of attributes of the nomophobia questionnaire that describe the students' sense of need to be with their smartphones. This component will consist of the following items: Q9, Q15, Q10, Q16, Q14 and Q8. They will be the factor that we call the "Anxiety Sensation" factor, which explains 38.668% of the total variability. The sense of anxiety of UNI students is related to the unfounded need for smartphones. This factor is basically caused by "not being able to communicate"; it refers to the feelings of losing instant communication with people and not being able to use the services that allow instant communication.

**Component 2:** This component contains six variables that are considered to be within those that do not find alternative resources to entertain themselves. This component will consist of the following items: Q12, Q6, Q13, Q5, Q11 and Q7. They will be the factor that we call "Compulsive Smartphone Use" factor, which explains 9.084% of the total variability, and which is reflected in the students compulsive need to interact with their smartphones. This factor is basically caused by "abuse and interference with other activities"; it refers to the excessive use of mobile devices even in situations where such use is dangerous or inconvenient.

**Component 3:** This includes the characteristics of low emotion. This component will consist of the following items: Q1, Q2, Q3 and Q4. They will be the factor that we call "Anxiety and Panic Sensation" factor, which explains 7.414% of the total variability. This factor reflects the mood of UNI students if they feel that they have been away from their smartphones for a long period of time. It is basically caused by "abstinence"; it refers to the withdrawal symptoms that appear if an addicted person cannot use the mobile.

The symptomatic nomophobia factors obtained in our research with engineering, sciences, and architecture students in Peru are very similar to those obtained in other studies cited by literature nomophobia with other students. For example, we mentioned some studies:

First, in [28], they found the following factors: problematic internet use, social appearance anxiety, and social media use. Second, in [16], they found the following factors: giving up convenience, fear due to running out of battery, nervousness due to disconnection from online identity, being uncomfortable when unable to stay up-to-date with social media, and anxious when unable to check e-mails. Third, in [23], they found the following factors: losing connectedness and giving up convenience. Then, in [14], they found the following factors: not being able to communicate, losing connectedness, not being able to access information, and giving up convenience. After that, in [42], they found the following factors: anxiety and depression. Finally, in [30], they found the following factors: abstinence, abuse and interference with other activities, tolerance, and lack of control.

This indicates that characteristic personality problems are common, regardless of the studies that are carried out and the region of the world where it occurs, because they are characteristic of human addiction. In university students of engineering, sciences, and architecture, the use of new technologies favors the academic training, the smartphone being one of them. The smartphone can be used as a self-learning tool; its benefits lead to excessive use resulting in nomophobia. This is a cause of concern for university teachers around the world.

In addition to the research cited in the literature, we find that none of them uses the focus group technique. This tool facilitated the ability for all the test questions evaluated to be answered completely by the students.

In Figure 2, we show in greater detail the structural model and testing results.

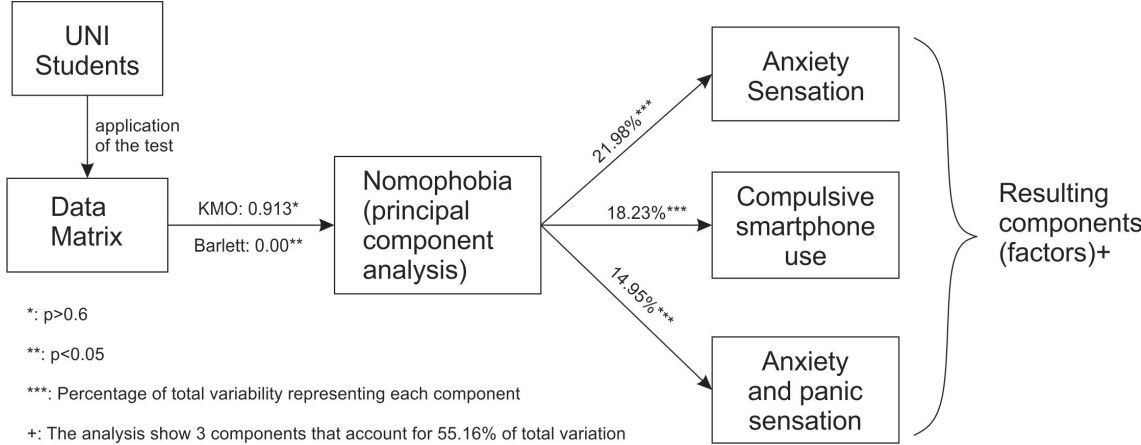

**Figure 2.** Structural model and results of applied analysis.

## 4. Conclusions

As a result of the process performed, following a research methodology adopted and validating the study, we obtain the following conclusions.

From the factor analysis, we concluded that there are three symptomatic factors of nomophobia in students at the National University of Engineering that describe the sensation that students experience when they feel the need to be with their smartphones and these are: sensation of anxiety, compulsive smartphone use, and sensation of anxiety and panic. These conclusions support our proposition that there is indeed enough evidence to state that there are symptomatic factors presented by UNI students due to indiscriminate use of smartphones. Based on the factors obtained in our study, we recommend implementing a program of prevention of levels of nomophobia that should be carried out by a medical center with specialists in the subject.

The use of a focus group allowed improving the quality of the data collected for research, and it was an innovative strategy to detect qualitative findings which improved the queries described in our questionnaire.

As future work, we hope to conduct research that links the influence of the symptoms of nomophobia factors, and the academic evaluation of students of the other universities, where the results obtained in our work would serve as a baseline for these research. Finally, we wish to expand the research to other social interest groups that will be necessary to have an instrument for the identification of symptomatic factors of nomophobia.

**Author Contributions:** J.A.R.-H. and J.L.C.-S. developed the ideas about the test adaptation, C.R.M.-A. designed the methodology, R.R.G.-L. implemented the questionnaire and validated the results obtained, E.E.A.-V. built the database and processed them. All of the authors were involved in preparing the manuscript.

**Funding:** The study was funded by the Vice-Rector for Research of the National University of Engineering.

**Conflicts of Interest:** The authors declare no conflict of interest.

## Abbreviations

The following abbreviations are used in this manuscript:

MDPI     Multidisciplinary Digital Publishing Institute
DOAJ     Directory of open access journals
TLA       Three letter acronym
LD        linear dichroism

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
