# Peer review of "Determining Symptomatic Factors of Nomophobia in Peruvian Students from the National University of Engineering"

_applsci, doi:10.3390/app9091814_

Round 1
Reviewer 1 Report
Male and female percentages are very different which may significantly effect the results.
The questions are not designed to measure the intend
Groups are designed to create homogeneity within (acting as blocks) so comparing them with Tukey' comparison is not appropriate. If that was not the purpose, it is not very clear following the analysis.
Author Response
In the word are the corrections

Reviewer 2 Report
Dear Authors:
I appreciated the opportunity to have your paper describing the project reviewed for possible publication in Applied Science. I have completed my review of the submit article and found that the article needs work.
The major area of concerns that limit my recommendation of the manuscript for publication is the review of literature, methodology, and conclusion. Although you made some efforts on those sections, they need to reframe the study that can clearly describe the extant literature on mental health challenges and how the proposed issue can be a bit more generalizable. The rationale of why the authors used that particular method has to be clearly explained. The conclusion is not completely dependent on the findings. That part seems to be insufficient reasoning and is hard to be reached by the current findings. The last paragraph has to expend more. In summary, the discourse is not tight and authors should not swap words for identical constructs. However, my primary request is to better explain how combining the measures is the proper method to use for predicting the final outcomes.
I hope that it is useful to you as the authors continue conducting their research. Because of the practical and conceptual limitations of this study, I do recommend that the authors take the time to revise their manuscript. Thank you!
Author Response
In the word are the corrections

Round 2
Reviewer 1 Report
I disagree with the statement: the results of this study can help educators understand students’ inclination to use their smartphones at all times.
This paper can be a baseline for such studies in sampling.
Author Response
In the article add your suggestion

Reviewer 2 Report
The revision has been much improved, compare to the original version. I thank you for all of your hard work. Still, it will be nice to proofread your language, particularly lines 74-76, lines 95-99, lines 109-111, and lines 126-140. I advise those portions can be proofread by a native English speaker before the editor finalize your manuscript.
Author Response
In the article add your suggestion
